# Exposure to heat stress leads to striking clone-specific nymph deformity in pea aphid

**Hawa Jahan**[1,2]* , **Mouhammad Shadi Khudr**[1], **Ali Arafeh**[3], **Reinmar Hager**[1]

**1** Faculty of Biology, Medicine and Health, Division of Evolution, Infection and Genomics, School of Biological Sciences, Manchester Academic Health Science Centre, The University of Manchester, Manchester, United Kingdom, **2** Faculty of Biological Sciences, Department of Zoology, University of Dhaka, Dhaka, Bangladesh, **3** Faculty of Science and Engineering, Chemical Engineering, James Chadwick Building, The University of Manchester, Manchester, United Kingdom

☙ These authors contributed equally to this work.
* hawa.jahan@postgrad.manchester.ac.uk

**Data Availability Statement:** The data underlying the results presented in the study are available from the [Figshare] repository via the following data link: [https://figshare.com/s/100808000e8d70fe036a].

## Abstract

Climatic changes, such as heatwaves, pose unprecedented challenges for insects, as escalated temperatures above the thermal optimum alter insect reproductive strategies and energy metabolism. While thermal stress responses have been reported in different insect species, thermo-induced developmental abnormalities in phloem-feeding pests are largely unknown. In this laboratory study, we raised two groups of first instar nymphs belonging to two clones of the pea aphid *Acyrthosiphon pisum*, on fava beans *Vicia faba*. The instars developed and then asexually reproduced under constant exposure to a sub-lethal heat-wave (27°C) for 14 days. Most mothers survived but their progenies showed abnormalities, as stillbirths and appendageless or weak nymphs with folded appendages were delivered. Clone N116 produced more deceased and appendageless embryos, contrary to N127, which produced fewer dead and more malformed premature embryos. Interestingly, the expression of the HSP70 and HSP83 genes differed in mothers between the clones. Moreover, noticeable changes in metabolism, *e.g.*, lipids, were also detected and that differed in response to stress. Deformed offspring production after heat exposure may be due to heat injury and differential HSP gene expression, but may also be indicative of a conflict between maternal and offspring fitness. Reproductive altruism might have occurred to ensure some of the genetically identical daughters survive. This is because maintaining homeostasis and complete embryogenesis could not be simultaneously fulfilled due to the high costs of stress. Our findings shine new light on pea aphid responses to heatwaves and merit further examination across different lineages and species.

## Introduction

Global climatic changes, primarily owing to anthropogenic effects [1], have recently been associated with heatwaves that may unprecedentedly occur sooner in early summer in Europe [2–5]. Exposure to high temperatures has complex effects on the metabolism and life-history traits, including the development, fecundity, and population dynamics of herbivorous insects

**Funding:** Hawa Jahan is supported by the Commonwealth Scholarship Commission in the UK. The funders had no role in study design, data collection and analysis, decision to publish, or preparation of the manuscript.

**Competing interests:** The authors have declared that no competing interests exist.

[1,6,7]. The negative impact of exposure to heatwaves may extend to result in serious developmental delay, reproductive malfunctioning, mortality, and even extinction of the insect population exposed to temperatures above optimum thresholds [1,8–10].

Aphids, small soft-bodied phloem-feeding insects, show species- and clone-specific thermal tolerance to abrupt changes in temperature [11]. Zhang et al. (2019) exposed five aphid species, including two clones of *Acyrthosiphon pisum* (Harris), to sub-lethal temperature (38˚C) and found distinct differences in their survivability, development time, and fecundity, with higher clone-specific mortality and thermotolerance reported in the exposed pea aphids [12]. Since aphid development, dispersal and reproduction are largely influenced by their sensitivity to thermal changes [13], maintaining a balance between survival, growth, and reproduction is quite challenging under stress and may lead to compromised fitness [14,15].

*Acyrthosiphon pisum*, an important crop pest [16] and eco-evolutionary model organism [17], shows impressive phenotypic plasticity, a phenomenon of producing variable phenotypes in response to environmental stress [18,19]. Metabolic differences may contribute to the contextual variability of reproductive, ontogenetic, and phenotypic plasticities across polymorphic aphid lineages such as green and pink morphs; these variations can occur within different contexts [20–25]. The optimal temperatures for *A. pisum* range from 20˚C to 25˚C, with upper limits up to 30˚C, depending on the geographic location and adaptability of aphid lineages [13]. Transgenerational effects following exposure to heat stress have been shown to carry over to subsequent generations further affecting offspring development and reproduction [26,27].

It is noteworthy that malnutrition may lead to depressed embryo development and embryo reabsorption [15,28], while embryo retention may also occur as a cost of increased densities in certain secondary endosymbiont communities in aged aphids [29]. However, these phenomena have not been reported in heat-stressed aphids. Exposure to thermal stress can induce upregulation of heat shock protein (HSP) genes (*e.g.*, Enders et al. 2015 [15]). Particularly, heat shock protein families HSP90 (including HSP83) and HSP70 are associated with repairing denatured proteins and maintaining homeostasis [30]. Impaired production of the HSP genes as well as histone acetyltransferase (HAT) p300/CBP, which is a transcriptional co-regulator, can cause serious embryonic defects [31] and developmental abnormalities [32] in *Drosophila melanogaster* (Meigen) and *A. pisum* [33,34].

Furthermore, Fourier Transform InfraRed spectroscopy (FTIR) is a user-friendly and affordable technique that is utilisable in measuring and understanding the metabolic changes that underpin organism responses to environmental stress [35,36]. FTIR is helpful for metabolomic profiling [24] and structural analysis [37] of insects and their secretions [38], but the application of FTIR to examine thermal stress responses in aphids is still understudied.

In this exploratory laboratory study, we reared two pea aphid clones on *Vicia faba* var *minor* Harz under thermal optimum (22˚C) and exposed them to sub-lethal thermal stress (27˚C), resembling a heatwave, for 14 days. Respective aphid first instars developed in each thermal regime until maturity and produced parthenogenetic offspring. We test the following hypotheses: 1) Survival of developing aphids is not affected by exposure to thermal stress with no signs of embryonic anomalies. 2) Aphid clones show universal phenotypic and molecular responses to thermal stress.

## Materials and methods

### Experimental setup

We established populations of two pea aphid clones (N116 [green] and N127 [pink]) from respective single parthenogenetic apterae. These aphids descended from samples provided by Imperial College London. The clones are of different biotypes, with N116 being more prolific

than N127 [39]. The stock aphids were raised on two-week-old fava bean plants (*Vicia faba* var. *minor* Harz) germinated and grown in a growth cabinet at 22˚C, 70% RH, and 16D: 8N cycle. The aphid clones were maintained for hundreds of generations in these conditions. We used individual meshed enclosures fitted with plastic pots (9cm x 9cm) filled with Levington Advance F2 (ICL©, UK), a steam-sterilised modular growing compost that contained nutrients at a medium level.

We applied two experimental conditions in terms of temperature (i) thermal optimum (22˚C) which was the control as described for the stock culture above, and (ii) heatwave (thermal stress) as a constant high temperature (27˚C), based on a pre-experiment pilot. The latter revealed that at 35˚C, plants started wilting after a few days of their germination before aphid introduction. At 30˚C, both survived plants and the aphids of both clones died within a few days after aphid introduction. At 28˚C, aphids survived but barely reproduced. Eventually, at 27˚C, both aphid clones survived and reproduced after the introduction of the respective clones, but only a few offspring remained alive after their emergence. We observed and counted unexpected premature or deformed instar nymphs, which was unusual and never reported before out of the aphid mother body.

In the experiment, we always used seven first instars for plant infestation and the instars were introduced to the plant two weeks following germination. We watered the enclosures every two days. There were 2 aphid clones X 2 environmental conditions X 15 replicates = 60 enclosures, see (S1 Fig in S1 File) for experimental design. The aphids, which survived the heat exposure, matured, and produced offspring, were sampled across the clones and the conditions at the end of the experiment and preserved at -80˚C for further assays to examine the expression pattern of HSP genes and to understand whether the aphid clones may show different metabolic fingerprinting in response to the heatwave.

## Statistical analysis

All statistical analyses were done in R ver. 4.0.4 [40]. First, using an Anova model (Type II), *car* package [41], we tested the total number of survived mothers (SM) in the microcosm on Day 14 as a function of (i) Thermal stress (two levels: 22˚C [thermal optimum, control], 27˚C [thermal stress]); (ii) Aphid clone (N116 [green], N127 [pink]), (iii) plant total dry weight (TDW) as a covariate, and (iv) the interactions of these predictors. Second, we counted the deformed and live nymphs on Day 14. Using a Manova model (Type II), we tested the binary response variable (deformed nymphs [DJ], live nymphs [LJ]) as a function of the predictors (i-iv). Third, using a Manova model (Type II), we tested the Differentially Expressed Genes (DEGs) of the aforementioned HSP genes (binary response variable [HSP70, HSP83]) as a function of the predictors (i-iii). The models were parsimonious as the non-significant predictors were removed. Each model was followed by a posthoc TUKEY test of pairwise comparisons; *emmeans* package [42].

## RNA extraction, quantification, and quality assessment

Total RNA was extracted using RNeasy Mini Kit (Qiagen©, UK) according to the manufacturer's protocol including an on-column DNase digestion step. The DNase digestion was done using an RNase-free DNase set (Qiagen©, UK) according to the manufacturer's protocol. Extracted RNA was quantified by Qubit® 3.0 Fluorometer using RNA Broad-Range (BR) Assay Kits (Invitrogen, Life Technologies©) according to the manufacturer's instructions. RNA purity was checked using a Nanodrop spectrometer (Thermo Scientific©, UK) loading 1 μL eluted RNA sample and the 260/280 and 260/230 absorbance ratios were recorded. RNA integrity was assessed by running the samples in 1.5% agarose gel and high quality DNA-free

RNA samples were used for gene expression analysis using Quantitative Reverse Transcription PCR (RT-qPCR).

## qPCR

RNA is reverse transcribed into complementary DNA (cDNA) using the QuantiTect Reverse Transcription kit (Qiagen©, UK) according to the manufacturer's instructions. A geNorm analysis [43] was conducted to determine suitable reference genes, SDHB and NADH, from a set of commonly used pea aphid reference genes [44], (S1 Table in S1 File), that exhibit stability across experimental conditions (treatment group, genotype). The reaction volume in each well for the qPCR was 25 µL, comprising of 12.5 µL Quantifast SYBR green PCR master mix (Qiagen©, UK), 2.5 µL (10µM) forward primer, 2.5 µL (10µM) reverse primer, and 7.5 µL sample cDNA diluted 1:50 in nuclease-free water. After mixing the reaction mixtures by flicking, samples were picofuged and were run on an AriaMx qPCR machine (Agilent©, USA) on the following protocol: 1 cycle of activation of HotStar Taq Plus DNA polymerase at 95°C for 5min; 40 cycles of cDNA strand dissociation at 95°C for 10sec followed by primer annealing and extension at 60°C for 30sec. Additionally, a melt curve analysis was run for each plate with an initial melt step at 95°C for 1 min, dropping to 55°C for 30sec, followed by a 0.5°C interval/s incremental increase from 55°C to 95°C and the cycle threshold (Ct) value was determined. All samples were run in duplicates.

All unknown sample template values were interpolated from an eight-point 4-fold serial dilution standard curve, run in each plate, constructed from pooled cDNA. The selection of candidate genes (HSP83 and HSP70) was made based on their function in stress response, development, and embryogenesis [45,46]. Primers for these genes were designed by PrimerBlast (NCBI), (S2 Table in S1 File). The reaction mixtures for qPCR were prepared and run on an AriaMx qPCR machine following the same protocol, as mentioned above. All standards and samples were run in duplicates. After the data collection, mean Ct values of duplicates were used to estimate the values of samples from the standard curve. To generate values of relative gene expression, sample values were normalised to reference gene values, obtained from the same cDNA dilution. To confirm the amplification of a single qPCR product, a differential melt curve analysis was used.

## FTIR

To establish the differences in chemical compositions in response to stress, FTIR spectra were obtained and compared between samples of the aphid clones reared at 22°C or 27°C, at the end of the experiment. Aphid mothers were individually sampled in separate tubes and three replicates per condition were used for the FTIR measurement. The samples were analysed using a Vertex 70/70v FTIR spectrometer with Opus ver. 7.5 (Bruker, Germany) at the Department of Chemical Engineering, UoM. Each aphid was placed on the device crystal and squashed gently immediately after removing it from a dry ice box to avoid any degradation. Background and aphid sample spectra were recorded at 4 cm$^{-1}$ resolution and 400–4000 cm$^{-1}$ range from 32 scans. Averages of three replicates spectra across conditions and aphid clones were analysed and compared. The comparison was made for functional groups/wavelengths related to the metabolism of carbohydrates (1000 cm$^{-1}$–1300 cm$^{-1}$), proteins (1400 cm$^{-1}$–1800 cm$^{-1}$), and lipids (2700 cm$^{-1}$–3000 cm$^{-1}$) [24,47].

## Results

### Aphid survival and production of deformed nymphs

The numbers of the survived mothers that developed from first instars, during 14 days, differed significantly across aphid clones ($F_{(1,58)}$ = 7.39, P = 0.009). Survival under the thermal

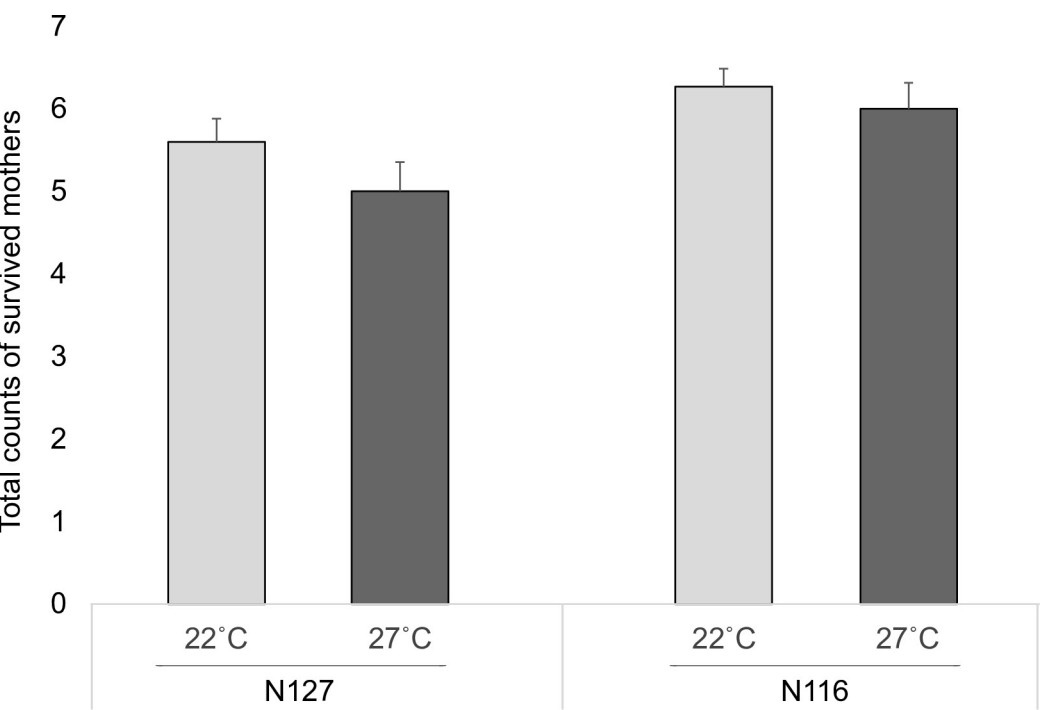

**Fig 1. Aphid survival.** The bars display the means (±SE) of the count of survived mothers at the end of the experiment on day 14, following their development from first instars. There were 60 enclosures in total including 2 aphid clones (N116 and N127) X 2 thermal conditions (control [22˚C] or severe [27˚C]) X 15 replicates.

optimum (control) was slightly higher than under thermal stress. More N116 mothers survived in both conditions when compared to N127 mothers, as mortality was higher under thermal stress in N127, (Fig 1).

Surviving mothers of the control produced live and healthy nymphs with functional appendages (Fig 2F and 2G). In contrast, subject to heatwave (thermal stress), surviving mothers produced striking scores of premature nymphs plus a few seemingly live but weak nymphs (Fig 2A–2E). The majority of the nymphs produced by N127 were with malfunctioned appendages (late embryonic stage); appendage-less dead nymphs (early embryonic stage) were also detected (Fig 2). However, all the deformed progeny of N116 were appendage-less dead nymphs from an early embryonic stage.

Only healthy nymphs were produced by both aphid clones in the control. Conversely, development under thermal stress not only resulted in a sharp decline in aphid population size but also the deformities documented in (Fig 2). For clone N127, the average number of live healthy nymphs in the control was ~26 times that recorded in the stressful environment. A similar pattern, yet more pronounced, was observed for the N116 clone, as the average count of live healthy nymphs was ~34 times that of the control compared to thermal stress. Furthermore, the average number of healthy nymphs produced in the control by the N116 mothers was ~2 times what the N127 mothers produced, indicating that the N116 clone was more prolific than N127 independent of any treatment. However, under thermal stress, the numbers of live and deformed nymphs were slightly higher in N116 than in N127 (Fig 3). Again, interestingly, there were no deformed or premature nymphs under the favourable hospitable conditions of the control (Fig 3).

The inferential stats revealed that thermal stress, aphid clone, and their interaction had highly significant effects on the numbers of both live and deformed nymphs ($F_{(1,51)}$ = 234.76, $P < 0.0001$), ($F_{(1,51)}$ = 19.04, $P < 0.0001$), and ($F_{(1,51)}$ = 18.31, $P < 0.0001$), respectively.

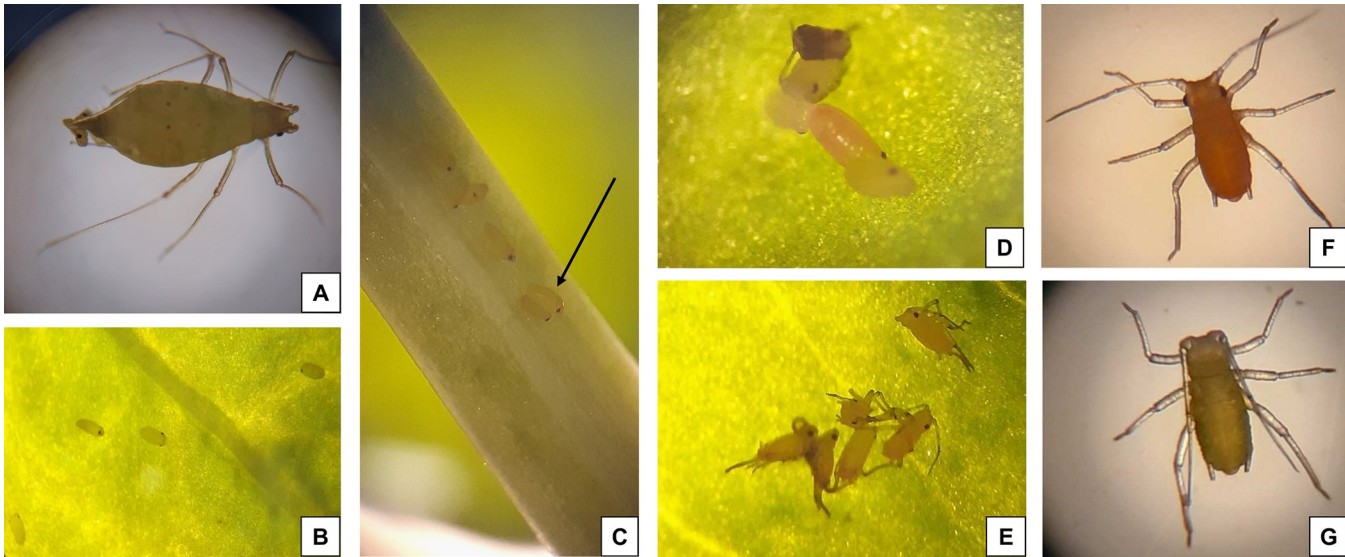

**Fig 2. Premature/Deformed nymph production under thermal stress.** The aphids of both clones that developed under 27˚C (heatwave) produced premature deformed nymphs and thrifty dull nymphs that generally appeared normal. Panel (A): an aphid mother (N116) producing a premature deformed nymph; Panel (B): some nymphs were born in early embryonic stages without any appendages (N116); Panel (C): An arrow pointing at a deformed nymph (N116) laid on the stem and lacking appendages; Panel (D): a deformed nymph (N127) with no appendages; Panel (E): some nymphs (N127) were born at their late embryonic stage but with folded (malfunctioning) appendages. In contrast, Panels (F) and (G) show close-ups of live healthy nymphs (N127 and N116, respectively) that were produced by aphids that developed under favourable temperate conditions (thermal optimum of 22˚C).

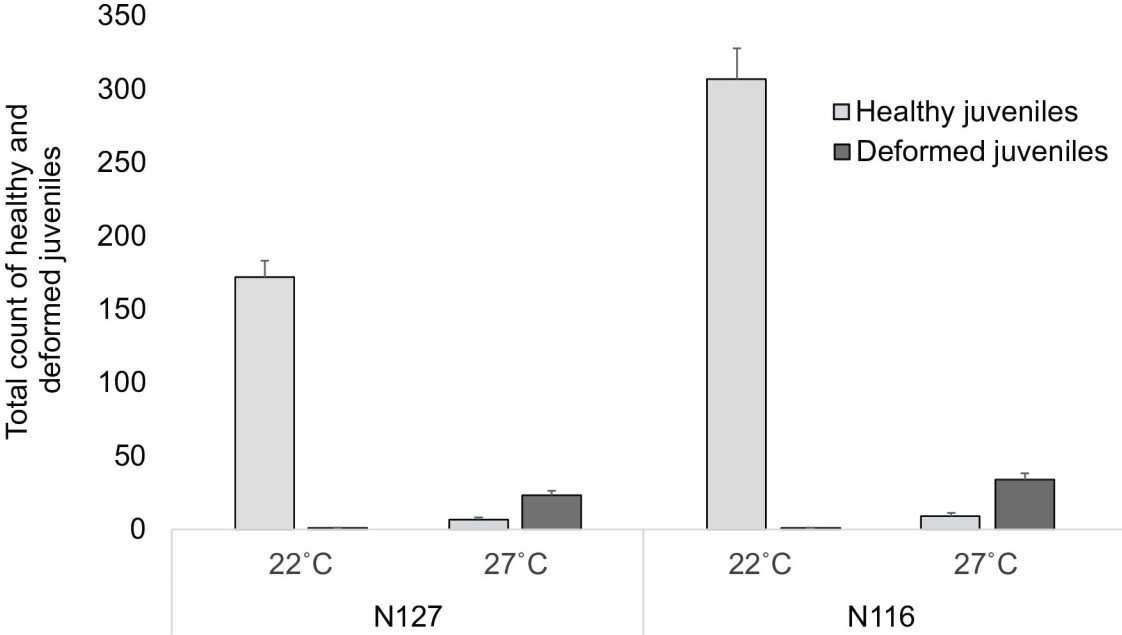

**Fig 3. Total numbers of nymphs.** The bars display the means (±SE) of the count of nymphs at the end of the experiment on day 14. Aphid mothers that developed and survived the heatwave produced healthy (light grey) and deformed (dark grey) nymphs. There were 60 enclosures in total including 2 aphid clones (N116 and N127) X 2 thermal conditions (control [22˚C] or severe, [27˚C]) X 15 replicates.

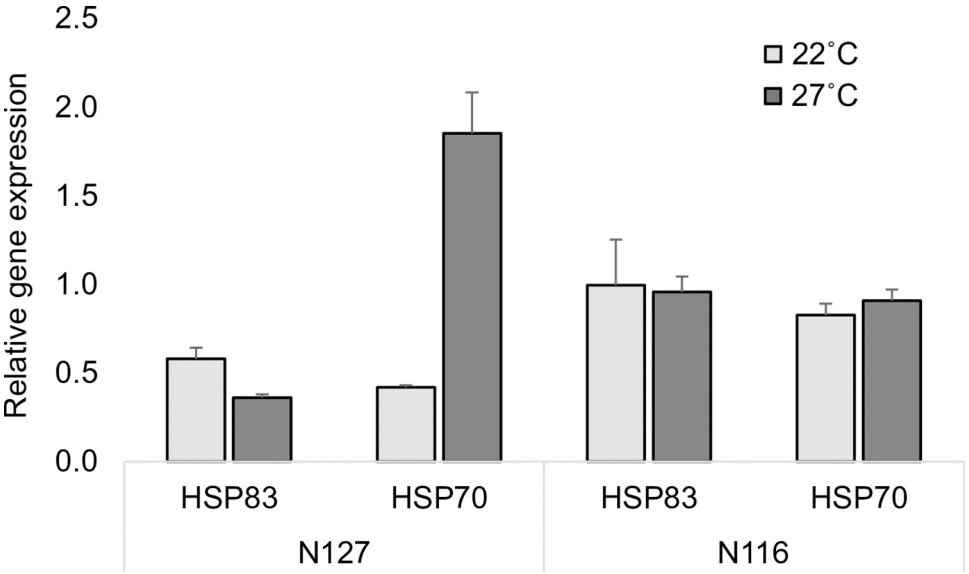

**Fig 4. Relative gene expression (HSP83 and HSP70).** The bars display the means (±SE) of relative gene expression of the aphid mothers in the control condition (light grey) and aphid mothers in the severe condition (dark grey) belonging to two aphid clones N116 and N127 at the end of the experiment. Gene expression was measured from three biological repeats consisting of pooled aphids from the respective genotypes across conditions, as specified in the Materials and methods.

Furthermore, TDW and the interaction (TDW X Thermal Stress X Aphid clone) had significant effects ($F_{(1,51)}$ = 4.02, P = 0.024) and ($F_{(1,52)}$ = 3.71, P = 0.031), respectively. Under thermal stress, TDW decreased slightly for N127-infested plants, while it considerably increased for the N116-infested plants, with the margin of difference in TDW between the thermal conditions being higher in the case of N116; see (S3 Table in S1 File) for full model details, and (S4 Table in S1 File) for TDWs. The posthoc TUKEY test revealed only the following pairwise comparisons as significant (N127–control *vs.* N127–severe, P < 0.0001), (N116–control *vs.* N116–severe, P < 0.0001), and (N116–control *vs.* N127–control, P < 0.0001).

## Heat Shock Protein (HSP) responses

HSP83 showed lower expression under severe environmental conditions (27˚C) for both clones, but the difference in comparison with the control (22˚C) was more apparent in N127, although the levels of expression were higher in N116 (Fig 4). Moreover, a higher HSP70 expression was detected under the severe conditions of 27˚C and that was more prominent in N127 than in N116. The inferential stats revealed that thermal stress ($F_{(1,7)}$ = 11.59, P = 0.006) and the interaction between aphid clone and thermal stress regime ($F_{(1,7)}$ = 8.69, P = 0.013) had significant effects on the expression of both heat shock protein genes (HSP70 and HSP83). The effect of aphid clone was marginally significant ($F_{(1,7)}$ = 3.76, P = 0.078). The posthoc TUKEY test only revealed the pairwise comparison (N127–control *vs.* N127–severe, P = 0.023) as significant; see (S5-S7 Tables in S1 File) for an alternative analysis perspective. As such, qPCR data showed significant differential expression in HSP83 and HSP70 genes in N127 (but not in N116).

## Metabolomic response

According to the FTIR spectra, lipids and fatty acids showed distinct higher intensities in the range (2849–2917 $cm^{-1}$) under thermal stress. A less prominent difference between the thermal conditions was seen at the wavelength (2957 $cm^{-1}$) (Figs 5 and 6). As for protein metabolism

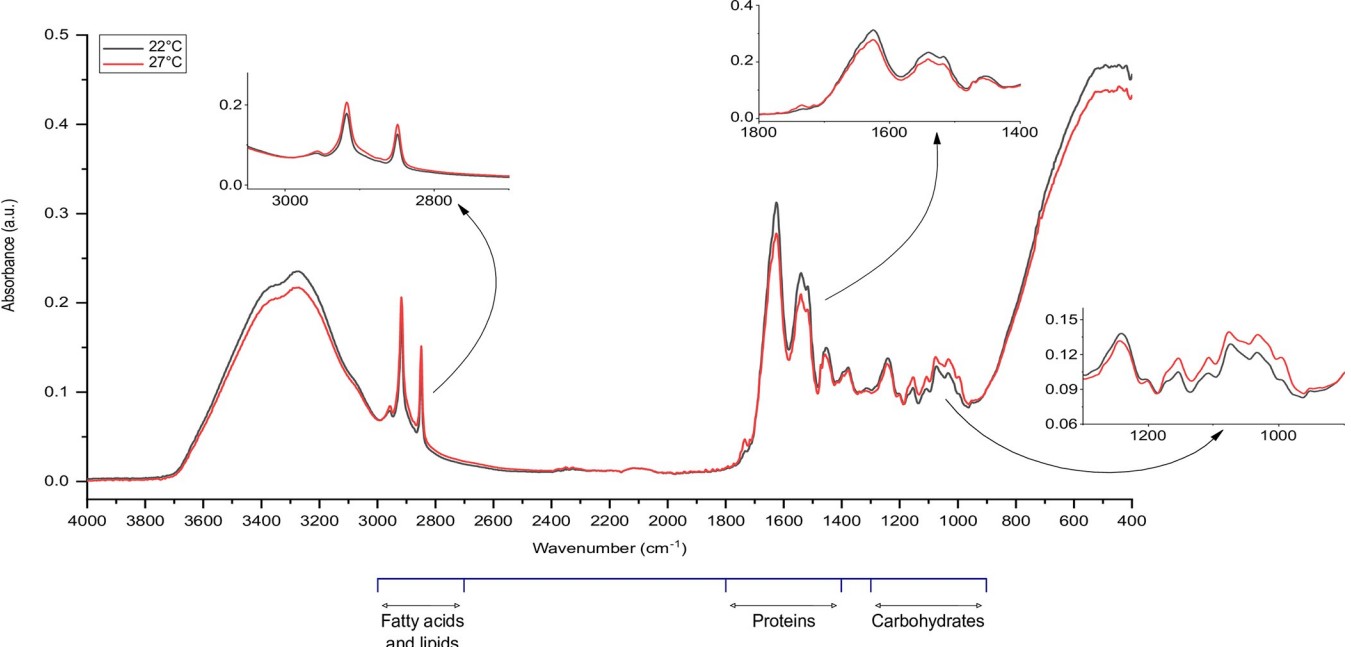

**Fig 5. FTIR spectra of thermally stressed pink pea aphid (N127).** The red line represents the spectrum for the aphids of the N127 clone raised under heat stress (27°C), while the black line represents the spectrum for the aphids raised at a favourable temperature (thermal optimum of 22°C). Metabolic changes (carbohydrates, proteins, fatty acids and lipids) of the aphid body were compared. The main spectroscopic regions are in the wavelength range of 4000–400 cm⁻¹.

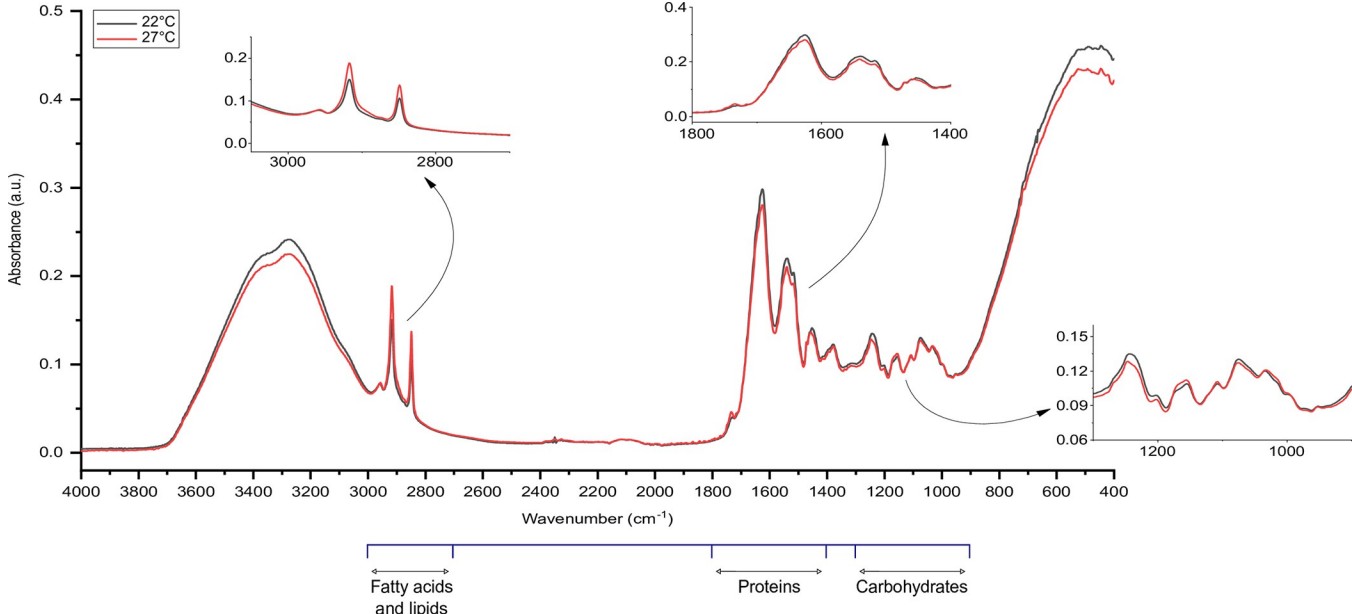

**Fig 6. FTIR spectra of thermally stressed green pea aphid (N116).** The red line represents the spectrum for the aphids of the N116 clone raised under heat stress (27°C), while the black line represents the spectrum for the aphids raised at a favourable temperature (thermal optimum of 22°C). Metabolic changes (carbohydrates, proteins, fatty acids and lipids) of the aphid body were compared. The main spectroscopic regions are in the wavelength range of 4000–400 cm⁻¹.

under thermal stress, distinct lower intensities within the range (1520–1627 cm$^{-1}$), whilst a higher intensity at (1736 cm$^{-1}$) were detected compared to the control. The patterns observed for the metabolism of lipids and proteins were generally identical for both aphid clones with slightly higher spikes for protein metabolism in N127 (Figs 5 and 6). Moreover, in the carbohydrate region, N116 showed a negligible difference between the two thermal conditions within the range (999–1154 cm$^{-1}$), but N127 had a tangible difference with higher intensities under thermal stress. However, comparative to the control, both clones displayed lower intensities at (1240 cm$^{-1}$), (Figs 5 and 6).

## Discussion

After 14-day exposure to thermal stress, the majority of aphid mothers of both clones survived but varyingly produced unexpected premature nymphs with deformities. These were either dead nymphs that lacked appendages or live deformed nymphs with folded malfunctioning appendages that died shortly. Whereas, the rest of the nymphs were weak and dull yet remained alive until the end of the experiment. We observed differential clone-specific expression of two HSP genes in the aphid mothers that developed, matured under stress, and survived compared to those reared in a thermal optimum. Additionally, certain metabolic changes were detected in response to thermal stress.

While the mortality rate observed in our study was lower than that reported by Enders et al. (2015) [15], concerning the survival of mothers, our findings highlight the harmful effects of constant thermal stress on aphid survival, which varies depending on the context and the clone being examined. The populations of the two clones dropped sharply whilst being exposed to the applied heatwave. This resonates with the findings of Chen et al. (2000) [48] on a reduced pea aphid fecundity at 25°C, and those of Dampc et al. (2021) [49] who found that exposure to 28°C affected the reproduction and longevity of rose aphid *Macrosiphum rosae* (Linnaeus) by altering plant-aphid interaction and their defence mechanism. Survival and nymph production in the severe condition were higher in N116 than in N127. That is attributable to a combination of factors including differences stemming from life-history traits [39], as well as stress responses on protein and metabolic levels, as shown by our findings and discussed below. More importantly, the exposure of the mothers to a constant sub-lethal heatwave in the present work led to developmental anomalies in the offspring. This incomplete or unsuccessful embryogenesis is quite an unusual phenomenon, which to the best of our knowledge, has never been reported before ex-ovaries. It is worthy of note that to date, only artificial knockdown of the genes HSP83 [33] and Lysine Acetyltransferase p300/CBP (HAT) [34] in pea aphid resulted in the formation of premature nymphs with folded appendages.

We discuss our findings from different vantage points. Firstly, our results lend support to those of Shingleton et al. (2003) [50] who examined extreme developmental regulation and the occurrence of embryonic malformation during diapause. A higher rate of developmental abnormalities is detectable when embryos develop at maintained perturbing high and/or fluctuating temperatures [50]. However, since diapause occurs at lower temperatures, it could be argued that aphids in our work experienced a state similar to quiescence that is usually associated with higher temperatures or abrupt and unexpected environmental change [51]. Given that the aphid clones here have been maintained in the lab for hundreds of generations where they never experienced heatwaves, the sudden change into unfavourable high sub-lethal temperature triggered a state similar to quiescence with a high risk of anomaly or physiological/developmental malfunction. The arrested/compromised development of some embryos [50], with a plausible state similar to quiescence, can be a compensatory mechanism that goes in line with interpreting fitness as a propensity [52] where only the fittest survive [52,53].

Secondly, aphids can rapidly respond to harsh conditions by preconditioning their developing offspring through extreme phenotypic plasticity (also known as polyphenism). This phenomenon allows them to exhibit swift adaptations to challenging environments [19,28,54]. However, a trade-off usually occurs where aphid mothers strive to balance between producing more energetically costly *alatae* to escape adversity and the default norm of producing *apterae* that are more fecund yet vulnerable to environmental risk [28,55]. In analogy, developing and maturing in a thermally challenging environment, most of the viviparous aphid mothers in the current work survived, but they produced a mix of dead, malformed, and few semi-healthy offspring. The sub-lethal heatwave partly impaired the aphid's ability to acclimate to constant stress exposure, affecting their balance between survival, development, and reproduction, and that compromised their coping mechanisms [56], but see [14]. Thus, a trade-off between survival and development might have taken place in the parthenogenetic all-female aphid populations, with arguably a plausible conflict between mothers and some of their developing embryos as well as between embryos [57,58] against nutrient deficiency resulting from critical changes in the density of their endosymbiont communities as we discuss below. It would be worth trying to entertain the prospect of sacrificing a portion of the developing daughters for the advantage of maintaining homeostasis in the bodies of their mothers. Aphids display exemplary cost-sensitive altruism against aphidophagous insects [59–62], but little is known on altruistic reproductive plasticity under severe physical conditions. If selective pressure favours altruism in harsh environmental conditions, as against natural enemies, future work is required to investigate a possibility of an altruism-orchestrating green-beard effect plausibly brought about by a nexus of genes associated with severe heat stress [58,63–65]. Even so, the question remains whether the observed phenomenon is an induced bet-hedging compensatory mechanism [66] under a thermally pressing ecological crunch or just a mere heat injury [67,68].

Thirdly, functional synthesis, mobilising, and conversion of metabolites are essential in maintaining energy homeostasis, embryogenesis, and stress resistance [69–71]. However, exposure to high temperatures may alter the functionality of these processes and result in deformities and compromised fitness as our findings suggest. Thermal stress depleted HSP83 expression in N127 but not in N116, which receives support from Will et al. (2017) [33] who found that reduced levels of HSP83 might lead to premature nymph development in aphid ovaries. HSP83 is a member of the HSP90 family of molecular chaperones that function as a determinant of fitness under non-optimal thermal conditions [14]. By contrast, HSP70 expression, which is usually upregulated under severe conditions [14], was higher under thermal stress in comparison with the control, with the highest level seen in N127. HSP70 is heavily involved in stress responses on many levels including synthesis, transcription, and metabolism [72,73]; elevated levels of HSP70, despite being essential to lessen stress burden, are energetically costly as it is ATP-dependent [72]. As such, the involvement of HSP genes in maintaining homeostasis, as well as anti-stress responses, may result in additional energy consumption to accommodate extra demands on resources, and that in turn may incur a fitness cost. There is evidence that an increased expression of HSP genes under thermal stress may negatively affect the natural development of embryos [50], leading to abnormal development, as shown in *D. melanogaster* [14]. Overall, the embryonic and larval defects examined in the current work could be the product of side effects of an increased upregulation of HSP70 for protection and maintenance as an anti-stress response, but our findings suggest that this may be clone-specific. The reported defects could also result from exhaustion or downregulation of HSP83 such that the expression of this HSP gene was insufficient to minimise embryonic anomalies and heat injuries in the exposed aphids.

As such, the aphids in our study may have faced a persistent challenge in balancing their allocation of resources between sustaining their typical life functions and defending themselves against harmful or disruptive heat injuries [7,15,74,75]. This dynamic created a trade-off resembling a war of attrition, where the aphids struggled to manage their limited resources to ensure survival. Further, the constant heatwave might have also aggravated the thermal challenge not only by impacting population dynamics [56] but also possibly by altering osmoregulation and water content in the insect body [76–79], as well as energy reserves [1,80]. There was a distinct rise in metabolomic fingerprints of lipids and fatty acids following exposure to thermal stress, which was consistent across clones. Lower reproductive rates under thermal stress may protect the accumulation and utilisation of saturated fatty acids and that can aid in alleviating the energetically costly effects of stress [81]. This is well documented under lower temperatures, *e.g.*, diapause [82], but consistent with the findings provided by Klepsatel et al. (2019) [83] on *D. melanogaster*, the described lipid metabolic response may be universal under low or high temperatures. Body fat is vital for functional metabolism and ovaries, and normal embryogenesis [68,82,84]. Thus, metabolic changes in body fat may reflect a possible adaptive stress response since insects must regulate the content and metabolism of their body fat [82,84–86] to offset stress-associated deficiencies [68,87,88]. All in all, mobilising and converting the storage of lipids and fatty acids can aid in counterbalancing the impacts of stress [89], corroborating our findings on active lipid metabolism in response to exposure to thermal disturbance. In a different light, conforming to the lipoid liberation theory [67,90], the constant exposure to non-lethal thermal stress in our study might have caused heat coagulation rather than protein denaturation that underlined embryonic deformities [67,90]. The coagulation might have compromised the essential roles of lipids in the haemolymph necessary for utilising energy, information signalling, and hence normal embryogenesis. Interestingly, thermal stress led to lower metabolic activity in the range (3000 cm$^{-1}$–3600 cm$^{-1}$), suggesting small changes in amides A and B regions (protein conformation) and membrane lipids [47].

Last but not least, aphid sensitivity to rising temperature and thermotolerance are associated with, or dependent on, their endosymbiont communities [12,48,91,92]. The density of the obligatory endosymbiont *Buchnera aphidicola* may vary according to aphid levels of maturity and exposure to thermal stress [93]. Insufficient availability of amino acids in the aphid diet due to lower densities of *B. aphidicola* in the thermally exposed hosts [12] could have also contributed to the production of deformed offspring in the current study. We note that N116 [62] and N127 harbour a range of facultative endosymbionts, with significantly lower titers in N127 (*Personal Observation*). This difference in the titre may contribute to mitigating the negative effects of stressors on aphids [94] and also influence clone-specific patterns of gene expression.

## Conclusions

The aphid clones in this study showed a continuum between ontogenetic plasticity, physiological change to resist stress, and failure to reproduce normally, although their populations did not go extinct. It was costly for the aphid mothers to survive in a constantly stressful thermal environment, as they incurred a fitness cost. Dead or premature and deformed offspring were produced due to compromised embryonic development. Aphid responses were shaped by the intertwined effects of endosymbiont density, life history, metabolic changes, differential expression of HSP genes, and a possible conflict between the survival of parthenogenetic mothers and the production of their offspring. This work casts light on a surprising response to atypical sub-lethal thermal stress and suggests a possibility for reproductive altruism, as it does for describing a new vulnerability of this important model organism and pest to heatwaves. Our findings carry significant implications for ecology, evolution, and agriculture. We

offer valuable insights that can be applied in future studies of eco-evolutionary dynamics, specifically in investigating organism responses to severe environmental changes. Furthermore, these insights can be instrumental in understanding the effects of abrupt environmental change and in developing more effective and innovative approaches to integrated pest management. Further research is necessary to examine whether this is unique to pea aphid or whether it may be more of a universal aphid anti-stress response.

## Supporting information

**S1 File. The supporting information file contains multiple supporting components including one supporting figure [S1 Fig. Experimental design] and six supporting tables [S1 Table].** List of reference genes tested to check expression stability; S2 Table. Details of two candidate genes (HSP70 and HSP83) used in qPCR; S3 Table. Analysis of the total number of nymphs (live and deformed); S4 Table. Total plant dry weight (TDW); S5 Table. Analysis of HSP70 expression; S6 Table. Posthoc test of HSP70 expression; S7 Table. Analysis of HSP83 expression].
(DOCX)

## Acknowledgments

We are thankful to Ms. Gemma Chapman and Dr. Abdullatif Alfutimie of the Chemical Engineering Department (CE), UoM, for their support at the FTIR facility therein.

## Author Contributions

**Conceptualization:** Hawa Jahan, Mouhammad Shadi Khudr.

**Data curation:** Hawa Jahan.

**Formal analysis:** Hawa Jahan, Mouhammad Shadi Khudr, Ali Arafeh.

**Funding acquisition:** Hawa Jahan.

**Investigation:** Hawa Jahan.

**Methodology:** Hawa Jahan, Mouhammad Shadi Khudr, Reinmar Hager.

**Project administration:** Mouhammad Shadi Khudr, Reinmar Hager.

**Resources:** Hawa Jahan, Reinmar Hager.

**Software:** Ali Arafeh.

**Supervision:** Reinmar Hager.

**Visualization:** Hawa Jahan, Mouhammad Shadi Khudr, Ali Arafeh.

**Writing – original draft:** Hawa Jahan, Mouhammad Shadi Khudr.

**Writing – review & editing:** Hawa Jahan, Mouhammad Shadi Khudr, Ali Arafeh, Reinmar Hager.

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
