## [Decision Letter · Decision Letter 0]

10 Apr 2023

PONE-D-23-04374Striking clone-specific nymph deformity in pea aphid exposed to heat stressPLOS ONE

Dear Dr. Jahan,

Thank you for submitting your manuscript to PLOS ONE. After careful consideration, we feel that it has merit but does not fully meet PLOS ONE’s publication criteria as it currently stands. Therefore, we invite you to submit a revised version of the manuscript that addresses the points raised during the review process.

We look forward to receiving your revised manuscript.

Kind regards,

Bilal Rasool, PhD

Academic Editor

PLOS ONE

“Hawa Jahan is supported by the Commonwealth Scholarship Commission in the UK.”

Additional Editor Comments:

Manuscript carried results in the text that needs revision for the improvement of the draft. Some suggested changes are in the comments portion to revise and improve the manuscript.

Reviewers' comments:

Reviewer's Responses to Questions

**Comments to the Author**

1. Is the manuscript technically sound, and do the data support the conclusions?

Reviewer #1: Yes

Reviewer #2: Yes

2. Has the statistical analysis been performed appropriately and rigorously? 

Reviewer #1: Yes

Reviewer #2: Yes

3. Have the authors made all data underlying the findings in their manuscript fully available?

Reviewer #1: Yes

Reviewer #2: No

4. Is the manuscript presented in an intelligible fashion and written in standard English?

Reviewer #1: Yes

Reviewer #2: No

5. Review Comments to the Author

Reviewer #1: The manuscript presents some very interesting results in the context of expression of two heat shock protein genes in pea aphid . The introduction have enough information about the cycle of Acyrthosiphon pisum; on its phenotypic variable in response to environmental stresses, with an adequate bibliography on past studies. The methodology is explicit in all of their parts. The discussion is clear and ensures that all findings are present

Reviewer #2: The manuscript lacks some important details to be fully understand. In addition, it is not well highlighted the importance of this study, and how the finds could be used in future. Please take into account that many sentences need to be rephrased, in my opinion. Some suggested changes as an example are in the comments portion to revise and improve the manuscript. The present form of draft required a lot of corrections. Please check for typos and inconsistencies in Journal style/formatting, Reference writing, abbreviations, scientific names, missing italics, double spaces, missing information, Authors’ instructions etc. Please find the comments and suggested corrections.

Title: Revise the title of the paper

Introduction: eco-evo” please write the complete words instead of abbreviations and or rephrase the sentence

Reproductive, ontogenetic, and phenotypic plasticities….morphs, underlined by

metabolic differences” please rephrase the sentence

Thermal stress induces up regulation of heat shock protein genes” Please rephrase the sentence

Harz)” please recheck

(Levington F2).” What is this please explain

Material and methods: (SM)” please explain the abbreviation

qPCR” please add suitable references in this portion

Ct” please explain the abbreviations where first used

Results: that developed from early instars” The authors studied early instars or first instar” please clear this in the writings

Fig 1. early instars or first instar or the results of which instar

Please also mention in the figure 22˚C and 27˚C instead of severe” which seems confusing for the readers” you may mention about the control in the table explanation

Same for the figure 3 and 4” please mention the temperature studied instead of control, severe, optimal etc and in the table explanation which temp is control, optimal and severe

Please replace the text in the draft (Fig 2 [Panels F and G]).” as (Fig 2, F and G).” (Fig 2, [Panels A-E]). Please also correct this in the whole manuscript

Please also correct in the text Panel (A): Panel (B): Panel (C): Panel (D): Panel (E): Panels (F) and (G)” as Fig 2 A, 2B, 2C… etc

Line 239: HSP responses” please explain what is HSP, abbreviations

Severe environment” please rephrase the words

Fig 6. FTIR spectra of green pea aphid (N116) in response to thermal stress. The red line

represents the spectrum for the aphids of the N127 clone raised under heat stress (27˚C), while the

black line represents the spectrum for the aphids raised at a favourable temperature (thermal

optimum of 22˚C). Metabolic changes (carbohydrates, proteins, fatty acids and lipids) of the aphid

body were compared. The main spectroscopic regions are in the wavelength range of 4000 – 400

cm-1.” please concise the title of the figure

Figure 1-6: Please concise the titles of the figures, these are explanation rather than the titles

Discussion: As for the survival of mothers, although the mortality rate in our study is less than that

reported by Enders et al. (2015) [15],” Please rephrase the sentence

of Chen et al. (2000)” Dampc et al. (2021)” those of Shingleton et al. (2003)” remove from the text as already mentioned reference number.

a phenomenon of extreme phenotypic plasticity (polyphenism)” rephrased the sentence

The sub lethal heatwave partly compromised aphid's ability to cope with (or acclimate to) the constant stress exposure, challenging the balance between survival, development, and reproduction” Please rephrase the sentence

reduced/depleted” please use proper words

it is possible that the aphids in our study……and reducing heat injuries” Please rephrase the sentence

Insufficient availability of amino acids in the aphid diet due to lower densities of

B. aphidicola……deformed offspring in the current study.” Is there any exploration of endosymbiont Buchnera aphidicola to find the reason.” Please mention in the supplementary tables (if necessary)

Table S3: Pillai, approx F, num Df, den Df and Pr” Please explain what are these abbreviations” Please write at the end of the table

6. PLOS authors have the option to publish the peer review history of their article (what does this mean?). If published, this will include your full peer review and any attached files.

Reviewer #1: No

Reviewer #2: No

---

## [Author Response · Author response to Decision Letter 0]

5 May 2023

Thank you for sending the reviewers’ comments, based on which we have amended our manuscript. Here, we provide a point-by-point rebuttal letter.

Editor comments

“Thank you for submitting your manuscript to PLOS ONE. After careful consideration, we feel that it has merit but does not fully meet PLOS ONE’s publication criteria as it currently stands. Therefore, we invite you to submit a revised version of the manuscript that addresses the points raised during the review process.”

Re: Thank you for your feedback that helped us improve our manuscript further. We will upload the point-by-point rebuttal letter 'Response to Reviewers', accompanied by a marked-up copy of our manuscript 'Revised Manuscript with Track Changes' that highlights changes made to the original version, as well as an unmarked version of our revised paper without tracked changes 'Manuscript'. We kindly note that we also provide here the lines associated with the applied changes in the said ‘Manuscript’. 

“If applicable, we recommend that you deposit your laboratory protocols in protocols.io to enhance the reproducibility of your results. Protocols.io assigns your protocol its own identifier (DOI) so that it can be cited independently in the future. For instructions see: https://journals.plos.org/plosone/s/submission-guidelines#loc-laboratory-protocols. Additionally, PLOS ONE offers an option for publishing peer-reviewed Lab Protocol articles, which describe protocols hosted on protocols.io. Read more information on sharing protocols at https://plos.org/protocols?utm_medium=editorial-email&utm_source=authorletters&utm_campaign=protocols.”

Re: Thank you, but we wish not to deposit the protocol because it is presented in the Methods. 

Journal requirements

“Please ensure that your manuscript meets PLOS ONE's style requirements, including those for file naming. The PLOS ONE style templates can be found at

https://journals.plos.org/plosone/s/file?id=ba62/PLOSOne_formatting_sample_title_authors_affiliations.pdf”

Re: Done as requested. We have carefully followed PLOS ONE’s guidelines while revising the manuscript. 

“Thank you for stating the following financial disclosure:

“Hawa Jahan is supported by the Commonwealth Scholarship Commission in the UK.”

Please include this amended Role of Funder statement in your cover letter; we will change the online submission form on your behalf.”

Re: We have amended the statement to read as follows: “Hawa Jahan is supported by the Commonwealth Scholarship Commission in the UK. The funders had no role in study design, data collection and analysis, decision to publish, or preparation of the manuscript.” The amended statement has now been added to the Funding statement and the cover letter. 

“Manuscript carried results in the text that needs revision for the improvement of the draft. Some suggested changes are in the comments portion to revise and improve the manuscript.”

Re: We have now revised the corresponding text in response to the comments, as specified below. 

“While revising your submission, please upload your figure files to the Preflight Analysis and Conversion Engine (PACE) digital diagnostic tool, https://pacev2.apexcovantage.com/. PACE helps ensure that figures meet PLOS requirements…. Please note that Supporting Information files do not need this step.”

Re: We have checked all the figures via PACE. as recommended. and applied any required adjustments accordingly. 

Reviewer Comments

“Have the authors made all data underlying the findings in their manuscript fully available?

The PLOS Data policy requires authors to make all data underlying the findings described in their manuscript fully available without restriction, with rare exception (please refer to the Data Availability Statement in the manuscript PDF file). The data should be provided as part of the manuscript or its supporting information, or deposited to a public repository. For example, in addition to summary statistics, the data points behind means, medians and variance measures should be available. If there are restrictions on publicly sharing data—e.g. participant privacy or use of data from a third party—those must be specified.”

Reviewer #1: Yes

Reviewer #2: No”

Re: All data presented in this work underlying our findings were made available in the original manuscript via the data repository Figshare URL therein, as we provided that data statement in the original and revised versions in compliance with PLOS ONE’s guidelines; see revised version (Lines 732 – 734). While, we appreciate that Reviewer #1 acknowledged that, it is not clear why Reviewer #2 stated otherwise without giving an example. 

“Is the manuscript presented in an intelligible fashion and written in standard English?

PLOS ONE does not copyedit accepted manuscripts, so the language in submitted articles must be clear, correct, and unambiguous. Any typographical or grammatical errors should be corrected at revision, so please note any specific errors here.”

Reviewer #1: Yes

Reviewer #2: No”

Re: Our manuscript is written in standard English, and we have now thoroughly revised it to accommodate the suggestions made by Reviewer #2 on improving the readability. 

REQUESTED REVISIONS:

Reviewer #1:

“The manuscript presents some very interesting results in the context of expression of two heat shock protein genes in pea aphid . The introduction have enough information about the cycle of Acyrthosiphon pisum; on its phenotypic variable in response to environmental stresses, with an adequate bibliography on past studies. The methodology is explicit in all of their parts. The discussion is clear and ensures that all findings are present.”

Re: Thank you very much indeed. 

Reviewer #2: 

Point 1: “The manuscript lacks some important details to be fully understand. In addition, it is not well highlighted the importance of this study, and how the finds could be used in future.”

Re: Thank you for your valuable feedback, which has helped us further improve our manuscript. We have thoroughly addressed all of your comments and provided a detailed response below. Additionally, we have revisited the Conclusions to emphasise the significance of our study and how our findings can be applied in the future, Lines 418 – 425. 

Point 2: “Please take into account that many sentences need to be rephrased, in my opinion. Some suggested changes as an example are in the comments portion to revise and improve the manuscript.”

Re: We are sorry to hear that. We have now revisited and amended the corresponding texts throughout the manuscript. Please, see our responses and/or rebuttal below. 

Point 3: “The present form of draft required a lot of corrections. Please check for typos and inconsistencies in Journal style/formatting, Reference writing, abbreviations, scientific names, missing italics, double spaces, missing information, Authors’ instructions etc. Please find the comments and suggested corrections.”

Re: Thank you. We have carefully as well as thoroughly re-checked the entire manuscript for any possible errors to correct. We can confirm that the manuscript is compliant with PLOS ONE’s guidelines on style/formatting, referencing (in-text and bibliography), abbreviations, scientific names, etc. See further details below. 

Title:

Point 4: “Revise the title of the paper”

Re: We have given the title another shot to refine and the revised one reads as follows: 

Exposure to heat stress leads to striking clone-specific nymph deformity in pea aphid 

Introduction:

Point 5: “eco-evo” please write the complete words instead of abbreviations and or rephrase the sentence”

Re: We have used ‘eco-evolutionary’ instead of “eco-evo” in the revised text, Line 70.

Point 6: “Reproductive, ontogenetic, and phenotypic plasticities….morphs, underlined by metabolic differences” please rephrase the sentence”

Re: Done, the revised text reads as follows: Metabolic differences may contribute to the contextual variability of reproductive, ontogenetic, and phenotypic plasticities across polymorphic aphid lineages such as green and pink morphs; these variations can occur within different contexts, Lines 72 – 74.

Point 7: “Thermal stress induces up regulation of heat shock protein genes” Please rephrase the sentence

Re: Done as suggested. However, we kindly note that “upregulation” is one word. The revised text reads as follows: Exposure to thermal stress can induce upregulation of heat shock protein (HSP) genes (e.g., Enders et al. 2015 [15]), Lines 82 – 83.

Point 8: “Harz)” please recheck”.

Re: We apologise for the misplaced bracket, which has been removed now, Line 95.

Point 9: “(Levington F2).” What is this please explain”

Re: Levington F2 is a steam-sterilised modular growing medium for plants, with F2 referring to a medium level of nutrients. This compost is commonly used in horticultural, botanical, and plant-insect interaction labs. We have added an explanation in the revised manuscript, Lines 108 – 109.

Material and methods:

Point 10: “(SM)” please explain the abbreviation”

Re: SM refers to the survived mothers. To prevent confusion, we have moved the abbreviation to immediately follow the term to which it refers, Line 129.

Point 11: “qPCR” please add suitable references in this portion”

Re: We followed the instructions and guidelines of the kit provider (Qiagen©, UK) for a commonplace standard qPCR procedure and we cite the said company in the text. That said, we have added supportive citations on the selection of the candidate genes, Lines 151, 153, and 167. 

Point 12: “Ct” please explain the abbreviations where first used”

Re: We did spell this acronym out in full in the original text, as Ct refers to cycle threshold, see the revised version Line 162. 

Results:

Point 13: “that developed from early instars” The authors studied early instars or first instar” please clear this in the writings”

Re: These terms are synonymous, but we opted now for “first instars” instead of “early instars” universally throughout the entire manuscript, Lines 191 and 197. 

Point 14: “Fig 1. early instars or first instar or the results of which instar”

Re: Sorted, see above, please. 

Point 15: “Please also mention in the figure 22˚C and 27˚C instead of severe” which seems confusing for the readers” you may mention about the control in the table explanation”

Re: Agreed, we have now applied the changes to Fig 1 as recommended. We clearly defined in the Methods that 22˚C is the control and 27˚C is severe, and the caption of Fig 1 follows that, as we accompany again the degree with the intended respective context. However, we have now added square brackets instead of commas for clarity. We applied that also for all relevant figures, as well, Lines 198 – 199, 240 – 241, and 253 – 254. See also our response to Point 16 below. 

Point 16: “Same for the figure 3 and 4” please mention the temperature studied instead of control, severe, optimal etc and in the table explanation which temp is control, optimal and severe”

Re: Done as suggested; see our response to Point 15 above. 

Point 17: “Please replace the text in the draft (Fig 2 [Panels F and G]).” as (Fig 2, F and G).” (Fig 2, [Panels A-E]). Please also correct this in the whole manuscript”

Re: Done as requested, Lines 201 and 202 – 203. 

Point 18: “Please also correct in the text Panel (A): Panel (B): Panel (C): Panel (D): Panel (E): Panels (F) and (G)” as Fig 2 A, 2B, 2C… etc.”

Re: It seems that the reviewer may have overlooked the fact that the comment pertains to the caption of Fig 2. As such, it may be overly repetitive to cite Fig 2A, Fig 2B, and so on, within the caption of Fig 2, but, please refer to our response to Point 17 above. 

Point 19: “Line 239: HSP responses” please explain what is HSP, abbreviations”

Re: This is a very commonplace acronym referring to Heat Shock Proteins. We think that the reviewer meant to spell it out in the mentioned text for emphasis. We have, therefore, rewritten the sub-heading to read as follows: Heat Shock Protein (HSP) responses, Line 252. We have also introduced the acronym defined earlier in the text, Line 83. 

Point 20: “Severe environment” please rephrase the words”

Re: The comment is not clear, nor is the opinion on why to replace that text. Anyway, we have now clarified the text in the revised version, Lines 253 – 255.

Point 21: “Fig 6. FTIR spectra of green pea aphid (N116) in response to thermal stress. The red line represents the spectrum for the aphids of the N127 clone raised under heat stress (27˚C), while the black line represents the spectrum for the aphids raised at a favourable temperature (thermal optimum of 22˚C). Metabolic changes (carbohydrates, proteins, fatty acids and lipids) of the aphid body were compared. The main spectroscopic regions are in the wavelength range of 4000 – 400 cm-1.” please concise the title of the figure”

Re: The titles of Figs 5 and 6 are almost identical, yet the reviewer’s comment only referred to Fig 6. 

The title: FTIR spectra of green pea aphid (N116) in response to thermal stress” is already concise. However, we did our best to compact that short title to read as follows: FTIR spectra of thermally stressed green pea aphid (N116) for Fig 6, and FTIR spectra of thermally stressed pink pea aphid (N127) for Fig 5, Lines 281 and 286.

Point 22: “Figure 1-6: Please concise the titles of the figures, these are explanation rather than the titles”.

Re: We were surprised by the reviewer's comment because we had already made considerable efforts to ensure that the titles were concise. Moreover, we followed the journal's guidelines, which state that figure titles should be "concise and descriptive": 

Fig 1. Aphid survival. It is a two-word title and thus impossible to make it shorter. 

Fig 2. Premature/deformed nymph production in pea aphid clones under thermal stress. This could be trimmed down slightly further, however, to read as follows: Premature/deformed nymph production under thermal stress. Line 215.

Fig 3. Total numbers of nymphs. This one is quite concise already. 

Fig 4. Relative gene expression (HSP83 and HSP70). Likewise. 

See our response for Point 21 above regarding shortening the titles of Fig 5 and Fig 6. 

Discussion: 

Point 23: “As for the survival of mothers, although the mortality rate in our study is less than that reported by Enders et al. (2015) [15],” Please rephrase the sentence”

Re: This phrase makes up half of the sentence, as the mentioned text is followed by a comma and the rest of the sentence in the original text: “As for the survival of mothers, although the mortality rate in our study is less than that reported by Enders et al. (2015) [15], our findings highlight the detrimental impact of constant thermal stress on aphid survival in a context-dependent and clone-specific manner.” 

Having said that, we have now rephrased the whole sentence to read as follows:

While the mortality rate observed in our study was lower than that reported by Enders et al. (2015) [15], concerning the survival of mothers, our findings highlight the harmful effects of constant thermal stress on aphid survival, which varies depending on the context and the clone being examined, Lines 300 – 303. 

Point 24: “of Chen et al. (2000)” Dampc et al. (2021)” those of Shingleton et al. (2003)” remove from the text as already mentioned reference number.”

Re: We respectfully disagree with the reviewer's suggestion to delete the name of the researcher or research team in certain cases, as this goes against the guidelines of PLOS ONE for that matter. While it is true that the guidelines make displaying the year optional, it is important to note that there are occasions where showing the name of the researcher or research team is necessary and should be included*. Therefore, we believe it is best to follow the guidelines while also considering the specific requirements of each case.

* Examples: 

https://doi.org/10.1371/journal.pone.0276390

https://doi.org/10.1371/journal.pone.0208370

https://doi.org/10.1371/journal.pone.0180663

Point 25: “a phenomenon of extreme phenotypic plasticity (polyphenism)” rephrased the sentence”

Re: We have rephrased the associated text so that the revised version reads as follows: 

Secondly, aphids can rapidly respond to harsh conditions by preconditioning their developing offspring through extreme phenotypic plasticity (also known as polyphenism). This phenomenon allows them to exhibit swift adaptations to challenging environments, Lines 330 – 332.

Point 26: “The sub lethal heatwave partly compromised aphid's ability to cope with (or acclimate to) the constant stress exposure, challenging the balance between survival, development, and reproduction” Please rephrase the sentence”

Re: Done, as requested, and the revised text reads as follows: 

The sub-lethal heatwave partly impaired the aphid's ability to acclimate to constant stress exposure, affecting their balance between survival, development, and reproduction, and that compromised their coping mechanisms, Lines 337 – 339.

Point 27: “reduced/depleted” please use proper words”

Re: Since the reviewer approved the use of the word "depleted" at the beginning of the sentence in the original manuscript, Line 344, we believe that the slash (/) used between the words later did not meet the reviewer's preference, although this punctuation mark was used in the reviewer’s comment in Point 3 above. Therefore, we have removed the slash and retained only one word (reduced). The words used in the sentence convey the intended meaning accurately, Line 357.

Point 28: “it is possible that the aphids in our study……and reducing heat injuries” Please rephrase the sentence”

Re: We have rephrased the text to read as follows: As such, the aphids in our study may have faced a persistent challenge in balancing their allocation of resources between sustaining their typical life functions and defending themselves against harmful or disruptive heat injuries [7,15,74,75]. This dynamic created a trade-off resembling a war of attrition, where the aphids struggled to manage their limited resources to ensure survival, Lines 375 – 378.

Point 29: “Insufficient availability of amino acids in the aphid diet due to lower densities of

B. aphidicola……deformed offspring in the current study.” Is there any exploration of endosymbiont Buchnera aphidicola to find the reason.” 

Re: As mentioned in the manuscript, Lines 80 – 81 and 400 – 405, and as the citations in the text for this matter indicate, Buchnera aphidicola is mainly sensitive to thermal stress, i.e., temperatures exceeding the range of its thermal optimum. Since aphids are highly dependent on this endosymbiont due to their diet deficiency, any changes in the titre of the endosymbiont community can lead to a significant negative impact on the aphid host. Moreover, B. aphidicola is contextually sensitive to changes in the physiology and ontogeny of the aphid host. The changes overserved and discussed in our work provide a supportive explanation, as well. 

Point 30: “Please mention in the supplementary tables (if necessary) 

Table S3: Pillai, approx F, num Df, den Df and Pr” Please explain what are these abbreviations” Please write at the end of the table.”

Re: Agreed, we have defined these acronyms at the end of the said table of supportive information. We have also spelled Sum Sq in full at the end of Table S5 as Sum of Squares. 

“PLOS authors have the option to publish the peer review history of their article…. If published, this will include your full peer review and any attached files.

”

Re: “no”. Thank you.

---

## [Decision Letter · Decision Letter 1]

15 Jun 2023

Exposure to heat stress leads to striking clone-specific nymph deformity in pea aphid

PONE-D-23-04374R1

Dear Dr. Jahan,

We’re pleased to inform you that your manuscript has been judged scientifically suitable for publication and will be formally accepted for publication once it meets all outstanding technical requirements.

Kind regards,

Bilal Rasool, PhD

Academic Editor

PLOS ONE

---

## [Editor Report · Acceptance letter]

22 Jun 2023

PONE-D-23-04374R1 

Exposure to heat stress leads to striking clone-specific nymph deformity in pea aphid 

Dear Dr. Jahan:

I'm pleased to inform you that your manuscript has been deemed suitable for publication in PLOS ONE. Congratulations! Your manuscript is now with our production department. 

Kind regards, 

on behalf of

Dr Bilal Rasool 

Academic Editor

PLOS ONE